# A Comparative Study of Hybrid Machine-Learning vs. Deep-Learning Approaches for *Varroa* Mite Detection and Counting

**DOI:** 10.3390/s25165075

**Published:** 2025-08-15

**Authors:** Amira Ghezal, Andreas König

**Affiliations:** Lehrstuhl Kognitive Integrierte Sensorsysteme, Fachbereich Elektrotechnik und Informationstechnik, RPTU Kaiserslautern-Landau, 67663 Kaiserslautern, Germany; akoenig@rptu.de

**Keywords:** *Varroa* mite detection, hyperspectral imaging, machine learning, deep learning, model comparison

## Abstract

This study presents a comparative evaluation of traditional machine-learning (ML) and deep-learning (DL) approaches for detecting and counting *Varroa destructor* mites in hyperspectral images. As *Varroa* infestations pose a serious threat to honeybee health, accurate and efficient detection methods are essential. The ML pipeline—based on Principal Component Analysis (PCA), k-Nearest Neighbors (kNN), and Support Vector Machine (SVM)—was previously published and achieved high performance (precision = 0.9983, recall = 0.9947), with training and inference completed in seconds on standard CPU hardware. In contrast, the DL approach, employing Faster R-CNN with ResNet-50 and ResNet-101 backbones, was fine-tuned on the same manually annotated images. Despite requiring GPU acceleration, longer training times, and presenting a reproducibility challenges, the deep-learning models achieved precision of 0.966 and 0.971, recall of 0.757 and 0.829, and F1-Score of 0.848 and 0.894 for ResNet-50 and ResNet-101, respectively. Qualitative results further demonstrate the robustness of the ML method under limited-data conditions. These findings highlight the differences between ML and DL approaches in resource-constrained scenarios and offer practical guidance for selecting suitable detection strategies.

## 1. Introduction

In many computer vision applications, deep-learning (DL) approaches, particularly convolutional neural networks (CNNs), achieve state-of-the-art performance when large, labeled datasets are available. However, in small-data scenarios—where the number of training samples and annotations is limited or computational resources are constrained, DL performance often suffers [1,2,3,4,5]. This limitation becomes critical in agricultural and ecological monitoring tasks, where collecting and annotating thousands of images is impractical. For instance, honey bee colonies face numerous stressors: pesticides [6,7], poor nutrition [8,9], pathogens [10], and climate change [11], and among these, the ectoparasitic mite *Varroa destructor* is widely considered the most destructive, weakening bees feeding on their fat bodies and vectoring viruses, often contributing to significant colony losses [12]. Timely monitoring of *Varroa destructor* is essential, as infestations above critical thresholds can cause irreversible damage [12]. The economic consequences of such damage are profound. A 2016/2017 study estimated national economic losses due to *Varroa* in Austria (EUR 32 million), Czechia (EUR 21 million), and North Macedonia (EUR 3 million) [13]. These losses were primarily attributed to colony mortality, with Austria losing EUR 318 per lost colony and EUR 168 per weak colony, Czechia losing EUR 209 and EUR 119, and North Macedonia losing EUR 120 and EUR 46 per lost and weak colony, respectively (COLOSS Survey, 2016/2017). In Australia, economic losses from a *Varroa* incursion could reach up to USD 1.31 billion over 30 years, dropping to USD 933 million in a containment scenario [14]. These figures underscore the importance of developing efficient detection and control systems to mitigate the economic damage caused by *Varroa* infestations.

A common non-invasive method places a hive board under the hive for 24 h to collect fallen mites; the resulting daily mite drop count serves as the key decision metric. If this count exceeds the recommended threshold, beekeepers intervene, typically with locally approved acaricides (e.g., formic acid in Germany) or alternative controls, in order to protect colony health [15]. Several prototype AI-based systems have been proposed to automate *Varroa* counting on hive boards, using traditional RGB images and CNNs or other lightweight architectures. For instance, Voudiotis et al. [16] developed an IoT-based *Varroa*-detection system using CNNs, comprising an end-node image-capture device, a cloud service for real-time inference, a data concentrator, and a mobile app for offline detection. Picek et al. [17] developed a system that uses a standard mobile phone camera to capture images of a beehive board, applying traditional computer vision methods, a CNN to detect *Varroa destructor* mites. Chazette et al. [18] proposed a system that scans incoming bees with a custom CNN to detect *Varroa destructor* infestation and, when identified, targets those bees with a laser for removal before they enter the hive. Bee localization is achieved using an implicit shape model, and classification relies on a custom CNN model. Bjerge et al. [19] introduced a portable video-based monitoring device, equipped with multispectral lighting and a camera positioned at the hive entrance. Shurischuster et al. [20] implemented a two-stage pipeline, first detecting bees at the hive entrance using an entrance camera, then identifying *Varroa* mites either via a sliding window approach with AlexNet or ResNet classifiers or by performing segmentation with DeepLabV3. Bilik et al. [21] applied object-detection techniques to RGB images based on YOLOv5 and SSD detectors for visual identification of *Varroa destructor*.

Moreover, Divasón et al. [22] introduced an open-source deep-learning pipeline for *Varroa* mite detection on hive boards using RGB images captured with smartphone cameras. Their approach employed a two-stage Faster R-CNN architecture, along with preprocessing techniques such as motion deblurring and image tiling, achieving high mAP scores on a relatively large annotated dataset. Although these efforts highlight the potential of deep-learning models, none have evaluated DL performance under severely limited data conditions. While this work [22] inspired our use of Faster R-CNN, our study differs by extending the problem to calibrated hyperspectral imaging, where richer spectral features can enhance object detection, and by investigating performance under limited dataset. n.

In our previous work [23] a classical pipeline (PCA→kNN→SVM) trained on just six hyperspectral images, achieved over 99% segmentation accuracy and high counting fidelity using just 6 labeled images. Here, we present a controlled comparison, classical ML versus deep learning (Faster R-CNN with ResNet-50/101 backbones), using the same six images, assessing standard detection metrics (precision, recall, F1-score, mAP@0.5) as well as computational aspects (training and inference time, hardware requirements) and visual output inspection.

## 2. Materials and Methods

### 2.1. Data Acquisition, and Calibration

All hyperspectral images used in this study are identical to those from our previous work [23]. Data were collected from the *Varroa* boards (mite-drop boards), which were obtained from the Bee lab hives of the IndusBee 4.0 activity in August 2022 and August 2023. The *Varroa* board functions as a passive collection surface placed beneath the hive, where fallen mites and debris accumulate over time. We utilized a Photonfocus MV1-D2048 × 1088-HS02-96-G2-10 hyperspectral camera [24] equipped with an IMEC snapshot-mosaic CMV2K-SSM5 × 5-NIR sensor, featuring a 5 × 5 mosaic of filters that yields narrow and contiguous 25 bands (675–975 nm) per frame. Each 2048 × 1088 pixel frame was acquired at up to 42 fps via GigE, allowing simultaneous capture of reflectance spectra for every pixel. For a full specification sheet, see Table 1 in [23]. The hyperspectral imaging was performed under controlled room temperature, using two 50 W halogen lamps (28 cm above the board at 45° angle) and Edmund Optics band-pass filters (675 nm LP, 975 nm SP, O.D. 4).

Each hyperspectral image consists of 2,228,224 pixels and covers an area of 1.85 cm × 3.55 cm, representing approximately 0.26% of the full *Varroa* board surface. For each pixel, a spectral profile of 25 bands was recorded and calibrated according to the protocol described in [23]:Reflectance calculation: . For each pixel, raw intensity values were converted to reflectance by subtracting the dark reference (to remove sensor dark current), then dividing by the difference between the white reference and the dark reference.Demosaicing: Each channel was demosaiced by linearly interpolating missing values along rows and columns, producing a complete reflectance data cube.Spectral correction: A correction matrix, built from IMEC supplied virtual-band coefficients, was used to adjust for spectral overlap between adjacent channels. Each pixel’s raw spectrum was multiplied by this matrix to obtain spectrally refined data.

The final calibrated 25-band reflectance data cube was saved in NumPy format, preserving both spectral (band) and spatial (x, y) dimensions. This ensured lossless storage and facilitated direct use in downstream analysis pipelines. To enable a fair and reproducible comparison, the exact same set of calibrated images was used for both machine-learning and deep-learning experiments. Although the hyperspectral imaging system provided high spectral resolution, maintaining data quality required strict acquisition protocols. Common challenges included non-uniform or variable illumination, sensor drift, and environmental fluctuations. To mitigate these effects, we maintained standardized lighting conditions throughout all acquisitions, which helped minimize confounding factors and ensured that observed differences in detection accuracy were primarily due to the models rather than inconsistencies in acquisition.

### 2.2. Annotation and Labeling

For both the ML and DL pipelines, six out of the twelve hyperspectral images were manually annotated to serve as training data. For the ML pipeline, pixel-wise segmentation was conducted using binary masks generated in GIMP tool. First, *Varroa* mite regions were meticulously outlined and filled in white, while the rest of the image (background and debris) was set to black. These masks enabled extraction of pixels corresponding to *Varroa* mites, which were labeled as class 1. To construct the background/non-*Varroa* class (class 2), a second set of masks was generated by manually highlighting selected regions of background and debris in white, with all other areas including *Varroa* mites, kept in black. Pixels extracted from these white, marked regions were labeled as class 2 (non-*Varroa*). From these masks, all pixels were extracted and labeled, yielding a total of 17,763 annotated instances (14,210 *Varroa* pixels and 3553 non-*Varroa* pixels). This approach provided well-defined, class-specific pixel samples for classification tasks. Additionally, for the shape based ML stage, all segmented candidate objects (white blobs) from eleven images were analyzed to extract geometric shape descriptors. For each candidate, eleven geometric features were computed using morphological analysis. Features were selected via Random Forest-based importance scores, and the final feature set comprised the six most informative shape characteristics including Area, Perimeter, Radius, Convex Hull Area, Solidity, and Circularity, for robust *Varroa*/non-*Varroa* discrimination. The extracted features from *Varroa* shapes are labeled as *Varroa* and other shapes feature values labeled as non-*Varroa*.

For the DL pipeline, tight rectangular bounding boxes were drawn around all visible mites, whether fully visible, partially occluded, or edge-cropped using the open-source LabelImg tool on the same six hyperspectral images used for training. Each bounding box was manually refined to closely fit the contours of each mite, based on visual inspection of the original hyperspectral images. Each bounding box was assigned the “*Varroa*” class label, with all non-mite regions left unlabeled and thus implicitly treated as background during training. Annotations were saved in the Pascal VOC XML format, which records bounding box coordinates and class labels for each image. All annotations were created by the author (with no external annotators involved). The training set comprised six images containing a total of 15 annotated mites, while six additional images, containing 38 mites in total, were reserved for validation and testing.

### 2.3. Data Splits and Sampling

Following acquisition, calibration, and annotation, each hyperspectral image (25 bands) and its corresponding Pascal VOC XML file were loaded and subjected to preprocessing. Per-band normalization was applied, for each of the 25 spectral channels, the mean and standard deviation were computed spatially and used to standardize the data, thus mitigating channel-wise illumination differences and supporting model convergence. All XML annotation files were parsed using a dedicated XmlParser class, which extracts bounding box coordinates, class labels, and matches each annotation to its specific image file. The parser validates and retrieves for each object the coordinates (xmin, ymin, xmax, ymax) and the associated class (“*Varroa*” or background). The VOC Dataset class, a custom subclass of torch.utils.data.Dataset, handled image loading, normalization, and bounding box/label parsing, returning tensors in the required format for Faster R-CNN input.

To enhance generalizability and model robustness, the training set underwent data augmentation using the Albumentations library, random horizontal flips and 90-degree rotations were applied to each image, while validation images received only format conversion to tensors. To gain insight into the deep feature representations learned by the models, patches corresponding to annotated *Varroa* and background regions were extracted from the six training images and passed through the trained ResNet-50 and ResNet-101 FPN backbones. From each patch, global feature vectors were formed by spatially averaging (mean pooling) the 256-dimensional outputs from each FPN pyramid level (P2–P5), yielding 1024-dimensional vectors. Principal Component Analysis (PCA) was then performed on these vectors: for ResNet-50, the first two components explained 99.3% of the variance; for ResNet-101, 99.8%. The resulting PCA plots (Figure 1a,b) visualize class separability in the learned deep embedding space.

For downstream analysis in ML pipeline, and to manage computational load, we implemented a two-stage data splitting and sampling procedure. First, 80% of the labeled pixel data were randomly selected to serve as the working dataset for both model training and testing. This subset was then partitioned into two splits: 10% of the data were designated for training, while the remaining 90% was allocated for testing and validation. All pixel spectra from the training split, each comprising 25 spectral bands, were subsequently subjected to Principal Component Analysis (PCA) for dimensionality reduction. The five principal components that captured approximately 93.6% of the total spectral variance were retained and used as the final input features for ML classification models. For shape-based classification, an 80/20 train-test split was applied to the 11-image dataset, with the larger portion used for training and the rest for model evaluation. A set of 6 geometric features was extracted from segmented blobs and labeled for further processing. Principal component analysis (PCA) was applied to the feature sets, raw pixel features for segmentation and 6-dimensional shape descriptors for shape detection, to visualize variance structure and separability. Silhouette scores were computed on the resulting embeddings to quantify cluster separation between *Varroa* and non-*Varroa* classes. Figure 1c,d illustrates the first two principal components of each feature set.

### 2.4. Software and Hardware Environment

All experiments were conducted using adopted open-source software libraries and executed on dedicated hardware tailored to the computational requirements of each pipeline. The ML pipeline was implemented in Python 3.9, leveraging scikit-learn 1.3.2 for model development and hyperparameter tuning, NumPy 1.26.2 for numerical operations, and OpenCV 4.10.0 for image processing. All ML experiments were performed on a local workstation equipped with an Intel Core i5-7500 CPU (4 cores, 3.4 GHz) and 16 GB RAM, requiring only CPU resources. Deep-learning experiments were conducted using PyTorch 2.5.1 and torchvision 0.20.1, with computations accelerated by CUDA 12.4. All DL training and inference ran on a remote Linux workstation featuring dual AMD EPYC 7262 8-core processors (16 physical cores, 32 threads at 3.2 GHz), 504 GB DDR4 RAM, and an NVIDIA Tesla V100-SXM2 Graphics Processing Unit (GPU 32 GB VRAM). Hyperparameter optimization including learning rate, weight decay, and scheduler step size was performed with Optuna’s Bayesian optimization framework. Data augmentation for DL training used Albumentations 1.3.0, incorporating horizontal flips and random rotations to enhance generalization.

Manual annotation of bounding boxes for the DL dataset was performed with LabelImg GUI v1.8.6. For the ML pipeline, binary mask annotation was conducted using GIMP 2.10.34. Data preprocessing and visualization—including PCA projections and confidence histograms—were performed with Matplotlib 3.9.4 and scikit-learn 1.3.2. Detection metrics such as mean Average Precision (mAP) were computed using the TorchMetrics package v0.11.4. For ML, hyperparameter searches were conducted via scikit-learn’s GridSearchCV. Table 1 summarizes the computational resources and execution times for each approach. The ML pipeline required only CPU resources, with a total training time of approximately 69 s and near-instant inference (∼8 milliseconds per image). In contrast, the deep-learning models required GPU acceleration, with each training session lasting over 17 min and average inference times of approximately 0.65 s per image.

## 3. Conducted Experiments

This section details the experimental procedures used to compare deep-learning (DL) and classical machine-learning (ML) approaches for *Varroa* mite detection and counting from hyperspectral imagery.

### 3.1. Deep Learning Pipeline

The proposed deep-learning detection pipeline comprises the following key components:Preprocessing: Data augmentation (random horizontal flips, rotations) was applied using Albumentations.Feature Extraction: The core model is Faster R-CNN with either a ResNet-50 or ResNet-101 backbone, modified to accept 25-channel input and equipped with a Feature Pyramid Network (FPN). The first convolution layer was reinitialized for compatibility with hyperspectral data.Region Proposal Network (RPN): The RPN generates anchor boxes and proposes candidate regions likely to contain *Varroa*.RoI Pooling and Head Network: Features corresponding to proposed regions are pooled and passed through classification and regression heads to predict class labels and bounding box coordinates.Post-processing: Non-maximum suppression (NMS) with IoU threshold 0.5 was applied to filter overlapping detections and generate the final set of bounding boxes and *Varroa* counts.

Both ResNet-50 and ResNet-101 backbones with Feature Pyramid Networks (FPN) were adapted to accept 25-band hyperspectral images by replacing the initial convolutional layer and reinitializing its weights. All other pre-trained weights were retained, except for the box predictor head, which was replaced to match the number of classes in this task (*Varroa* vs. background). Annotated hyperspectral images and corresponding Pascal VOC XML files were parsed, extracting bounding box coordinates and class labels. Images with missing channels, annotation inconsistencies, or shape errors were excluded. The dataset comprised 12 fully calibrated hyperspectral cubes of size 1088×2048×25 (pixels × pixels × bands), split at the image level: six images (with 15 annotated *Varroa* mites) were designated for training, and six (with 31 mites) for validation and final testing. This division matched the data used in the ML pipeline to ensure a fair comparison. Prior to training, each spectral band of every image was normalized (mean subtraction, division by standard deviation) to ensure comparable feature scales during model learning. During training, data augmentation was performed using Albumentations, random horizontal flips, and $90^{\circ }$ rotations, each with p=0.5, were applied to increase model robustness. Validation and test sets underwent only conversion to PyTorch tensors. For optimization, AdamW was used with an initial learning rate of $1\times 10^{-5}$ and weight decay $2\times 10^{-4}$ (ResNet-50) and $1\times 10^{-5}$/$5\times 10^{-6}$ (ResNet-101). A StepLR scheduler (step size 50 epochs, gamma 0.9) controlled the learning rate, and mixed precision training was enabled with GradScaler for efficiency. Hyperparameters, including learning rate, weight decay, batch size, and scheduler parameters, were tuned using Optuna [25], running 20 trials per backbone. For each backbone (ResNet-50 + FPN and ResNet-101 + FPN), 20 trials were run—each training for five epochs to minimize total loss for finetuning parameters. The best configuration was then used to train for 300 epochs with a batch size of 1 and early stopping based on the minimum validation loss. Key training hyperparameters are summarized in Table 2.

Figure 2 shows the evolution of the training loss for both ResNet-50 + FPN and ResNet-101 + FPN architectures. In both cases, the loss decreases rapidly during the initial epochs and then stabilizes after approximately epoch 150–200. Based on this convergence behavior, we selected 300 training epochs to ensure thorough model optimization.

The region proposal network (RPN) used anchor sizes of (32, 64, 128, 256, 512) and aspect ratios of (0.5, 1.0, 2.0). At inference, predictions were filtered with a default confidence threshold of 0.5 and non-maximum suppression (NMS) IoU threshold of 0.5.

The DL pipeline, illustrated in Figure 3, employs a modern object-detection architecture (Faster R-CNN) adapted for hyperspectral imagery:

### 3.2. Machine Learning Pipeline

The machine-learning (ML) pipeline was designed to leverage both spectral and geometric information for *Varroa* mite detection, as summarized below:Input and Annotation: Each hyperspectral cubes are manually annotated with binary masks distinguishing *Varroa* from background pixels.Feature Extraction: From each annotated pixel in the six training images, the 25-band spectral signature was extracted, resulting in a high-dimensional data matrix.Dimensionality Reduction: Principal Component Analysis (PCA) was performed on the pixel-wise spectra, retaining the top five principal components.Pixel-wise Segmentation: The reduced feature matrix was classified using a 1-nearest-neighbor (1-NN) classifier, assigning each pixel to *Varroa* or non-*Varroa* class.Blob Generation and Shape Feature Extraction: Segmented images were used to identify blobs (connected *Varroa* regions), from which eleven geometric shape descriptors were computed for each blob.Feature Selection and Shape Classification: Six shape features identified, and a linear SVM (C = 1.0) was then trained to distinguish true *Varroa* blobs from false positives, providing the final *Varroa* count per image.

The same six calibrated images used for DL training were used in the ML pipeline to ensure fair comparison. For spectral analysis, the 25-band pixel vectors were reduced to five principal components, explaining ∼93.6% of the variance. A random 80% subset of all labeled pixels was selected and then split, with 10% for training and 90% for testing, to manage computational load. To improve computational efficiency, input images were resized from (1088, 2048) to (200, 400) using OpenCV interpolation prior to kNN-based segmentation. Post-classification, segmentation results were refined using a bilateral filter (kernel size = 9, sigma values = 75) to enhance *Varroa* shape preservation while reducing noise. Segmentation yielded 92 *Varroa* candidate blobs for training and 26 for testing(80%/20% split) was kept for shape analysis classification, serving as a targeted evaluation of the model’s *Varroa* counting ability, separated from the overall classification accuracy assessment on the remaining test set to determining the model’s performance. Eleven shape features were extracted from each blob, with the six most discriminative selected by Random Forest feature importance (area, perimeter, radius, convex hull area, solidity, circularity). Multiple classical classifiers (linear SVM, RBF SVM, kNN, Random Forest, Decision Tree, and ANN) were evaluated via five-fold stratified cross-validation. The 1-NN classifier gave the best pixel-wise segmentation accuracy and was selected; a linear SVM was chosen for blob classification due to perfect held-out performance.

Figure 4 illustrates the full ML pipeline, while Table 3 details feature dimensions at each stage, allowing for side-by-side comparison with the DL approach. For both pipelines, we evaluated precision, recall, and F1-score; for DL, we also report mAP@0.5. Final *Varroa* counts were compared against ground truth in the test images.

## 4. Results

This section presents a comprehensive analysis of the comparative performance of machine-learning (ML) and deep-learning (DL) pipelines, covering quantitative benchmarks, and qualitative visualizations.

### 4.1. Quantitative Evaluation

We evaluated all pipelines at a standard IoU threshold of 0.50, reporting Precision, Recall, and F1-Score. Results are given in Table 4.

Table 4 summarizes the detection performance of both pipelines. Our ML baseline (PCA→kNN→SVM) correctly detected 37 out of 38 mites, yielding Precision = 0.9983, Recall = 0.9947, and F1-Score = 0.9918.

ResNet-50 + FPN detected 28 of 38 mites, achieving Precision = 0.966, Recall = 0.757, and F1-Score = 0.848. In contrast, ResNet-101 + FPN detected 30 of 38 mites, with Precision = 0.971, Recall = 0.829, and F1-Score = 0.894. Table 5 summarizes the main performance metrics across three different random seeds (42, 99, 123) for both backbone models. While some variability in the metrics is observed, both architectures demonstrate consistent performance, with relatively low standard deviations for each metric.

### 4.2. Visual Comparison of Detection Results

Figure 5 presents a qualitative comparison of detection outputs from the ML pipeline and both Faster R-CNN variants on six representative test images. In each image, for Faster R-CNN models, true positives (TP) are outlined in green (predictions overlapping a ground-truth mite), false positives (FP) in red (detections with no corresponding annotation), and false negatives (FN) in blue (annotated mites that were missed). The ML-based pipeline detected blobs are outlined in red, with correctly labeled *Varroa* overlaid in green.

Manual counting of the six test images (3, 6, 6, 7, 7, and 9 mites per image, respectively) yielded a total of 38 mites.

Table 6 lists, for each of the six images, the ground-truth (GT) mite count and the number of TP/FP/FN detected by each pipeline.

Figure 6 displays the distribution of detection confidence scores for true positives versus false positives on the ResNet-50 and resnet-101 models across all six images. Green bars indicate TP confidences; red bars indicate FP confidences and the blue bars indicate duplicate TP.

### 4.3. Hyperparameter Sensitivity Analysis

To systematically investigate the impact of key hyperparameters on model performance, we conducted a sensitivity analysis across both deep-learning (DL) and machine-learning (ML) pipelines. For the DL experiments, mean Average Precision (mAP) was computed after five training epochs for each trial. Figure 7 illustrates the relationship between each hyperparameter, learning rate, weight decay, batch size, step size, and gamma, and the corresponding mAP score for Faster R-CNN models using ResNet-50 and ResNet-101 backbones.

A similar sensitivity analysis was conducted for the ML pipeline using grid search with cross-validation. Specifically, the number of neighbors *k* (k∈1,3,5) in k-Nearest Neighbors (kNN) and the regularization constant *C* (C∈0.1,1,10) in Support Vector Machines (SVM) were tuned, and their effect on classification accuracy was visualized. The results are shown in Figure 8.

Across both deep-learning and classical ML models, the analyses show that performance is highly sensitive to hyperparameter choices. In the DL pipeline, Faster R-CNN exhibited notable variation in mAP depending on learning rate, weight decay, and scheduler settings (see Figure 7). The implications of these findings are further discussed in Section 5.

## 5. Discussion

Inspection of the training loss curve (Figure 2) reveals that model optimization proceeded efficiently, with substantial loss reduction occurring within the first 150 epochs. After this point, the loss remained stably low, suggesting that additional training beyond epoch 150 provided limited further improvement. This stabilization supports the choice of 300 training epochs as sufficient to ensure convergence and avoid underfitting, while also demonstrating that early stopping could be considered in future work to optimize training time.

Our experiments reveal a clear comparison between the classical ML pipeline and the two ResNet + FPN detectors.

In the ML pipeline, the PCA projections of raw spectral pixel features (Figure 1a) and 6-D blob shape descriptors (Figure 1b) exhibit clear separation between *Varroa* and non-*Varroa* patches in the first two principal components. For the deep networks, we concatenate the 256-channel feature maps from FPN levels P2–P5 into a single 1024-dimensional vector, apply global average pooling, and project via PCA. In comparison, raw-pixel ML features achieve a silhouette score of 0.74, and shape descriptors explain 65.2% of the variance with a silhouette score of 0.68, indicating reasonable morphological clustering. In contrast, ResNet-50 + FPN and ResNet-101 + FPN explain 99.0% (silhouette = 0.802) and 99.6% (silhouette = 0.848) of the variance, indicating a strong clustering [26]. Moreover, all silhouette scores exceed 0.5, confirming that deeper residual backbones produce very similar but distinctive features that enable near-linear separation of *Varroa* vs. non-*Varroa* clusters and yield well-separated feature groups. Despite the clear discriminative structure revealed by PCA and the high silhouette scores, both Faster R-CNN variants still underperform the simple ML pipeline on detection metrics (Table 4). Precision and F1-score decline from 0.998/0.992 for PCA + SVM + kNN to 0.966/0.848 for ResNet-50 + FPN and 0.971/0.894 for ResNet-101 + FPN, most likely a consequence of our limited training set: deep architectures have millions of parameters and, in data-scarce regimes, tend to overfitting and subsequent misclassification errors [5]. Qualitatively, the ML pipeline remains remarkably precise, correctly detecting all 38 visible mites except for a single mite partially occluded at the image boundary (Figure 5), this limitation previously noted in [23]. In contrast, both deep models exhibit inconsistent performance across images:ResNet-50 + FPN registered false negatives in Image1 (2TP/1FN), Image2 (4TP/2FN), Image4 (3TP/3FN), and Image5 (5TP/2FN), and produced one false positive in Image3 (7TP/1FP vs. 6GT).ResNet-101 + FPN missed three mites in Image4 (3TP/3FN) and one in Image5 (6TP/1FN), and detected only 7 of 9 mites in Image6 (7TP/2FN).

Overall, ResNet-50 yielded 28 correct detections, while ResNet-101 achieved 30 correct detections, both underperforming the ML approach’s 37/38 count. These results underscore that, despite their powerful representational capacity, deep models under limited training conditions can suffer from overfitting, leading to both missed detections and spurious classifications.

Furthermore, the Faster R-CNN variants exhibit counting errors due to the duplication in the ResNet-101 model. In Images 1, 3, 5, and 6, ResNet-101 generated duplicate true positives (each counted as a separate detection; see Figure 5). Resource efficiency represents another critical advantage of the ML pipeline.

As shown in Table 1, this stability incurs a steep computational cost: the classical ML pipeline trains in under 70 s and infers in 0.008 s per image on CPU, whereas both Faster R-CNN variants require over 1000 s of GPU training time and 0.65–0.67 s per image at inference. With inference times under 10 ms per image on CPU-only hardware, this approach is ideally suited for real-time, field-deployed *Varroa* monitoring. Conversely, the deep-learning models require GPU-based resources and considerably longer training and inference durations, constraining their use in mobile or edge-device environments.

The hyperparameter sensitivity analyses presented in Figure 7 and Figure 8 demonstrate that model performance is highly dependent on hyperparameter selection across both deep-learning and machine-learning pipelines. Only a narrow subset of configurations yielded optimal results, as measured by mean Average Precision (mAP) or classification accuracy, while most settings led to significantly degraded performance. For Faster R-CNN with a ResNet-50 backbone (Figure 7a), we observed a relatively broad distribution of good performing trials. Learning rate and weight decay, showed tight clustering values, with performance dropping sharply outside these zones, highlighting the importance of precision in tuning parameters. In contrast, ResNet-101 (Figure 7b) produced trials, with the best configuration achieving a low mAP value. This performance gap likely come from the model’s greater depth and complexity, which can make it more sensitive to suboptimal hyperparameters. The optimal configuration differed substantially: ResNet-50 favored lr = 1.46 × 10^−4^ and weight decay = 5.31 × 10^−4^, while ResNet-101 required lr = 4.74 × 10^−5^ and weight decay = 3.55 × 10^−5^, among other differences (e.g., step size = 100 vs. 50). These findings also suggest that deeper architectures may require longer training or more fine-grained searches to reach convergence. Other hyperparameters, such as batch size, step size, and gamma, exhibited non-linear influences on performance. Notably, optimal settings were often found at the edges of tested ranges (e.g., batch size = 1, gamma = 0.5), indicating that sensitivity is not evenly distributed in parameter space. The results in Figure 8 echo these findings within the classical ML pipeline. For k-Nearest Neighbors (kNN), performance peaked sharply at k=1, suggesting that larger neighborhoods introduce over-smoothing in high-dimensional or tightly clustered hyperspectral data. For Support Vector Machines (SVM), accuracy increased with the regularization parameter *C*, stabilizing beyond C=1, consistent with the advantages of allowing fewer margin violations in linearly separable data. Together, these observations underscore the necessity of automated hyperparameter tuning frameworks. Manual selection, especially for deep-learning models, is not only inefficient but also prone to overlooking optimal configurations. Future work may extend these efforts using advanced AutoML platforms, which allowed us to efficiently identify high-performing configurations while minimizing computational overhead. This not only improved model performance but also enhanced reproducibility and reduced reliance on manual tuning. Such systems can automate not just hyperparameter search but also architecture optimization, model selection, and training workflows. This will be especially valuable in real-world deployments where datasets evolve and expert intervention is limited.

The stabilization of training loss observed after epoch 150–200 for both backbones (Figure 2) confirms that the selected training schedule was effective, with both models achieving convergence well before 300 epochs. This justifies our choice of training duration, extending training further would yield minimal improvement while unnecessarily increasing computational cost. The results also highlight the adequacy of our chosen hyperparameters and learning rate schedule for this work.

Despite careful efforts to control randomness by setting random seeds and fixing hyperparameters, some variability remains in the results. This is a well-known limitation of deep learning, stemming from inherent non-deterministic operations in GPU-accelerated libraries such as CUDA and cuDNN [27]. As a result, retraining the same model with the same configuration can lead to minor differences in metrics or training curves. To address this, we report the mean and standard deviation across several runs (Table 5), which provides a more robust estimate of model performance than a single run. While perfect reproducibility is not currently feasible for deep-learning models, the low standard deviations observed here confirm the robustness of our reported results. In contrast, classical machine-learning algorithms—such as KNN and SVM—produced fully reproducible results in our experiments, repeated runs with the same data splits always yielded identical metrics. This finding further highlights the reproducibility gap between traditional ML and deep-learning approaches.

Full-Board Image Processing and Future DirectionsAn ongoing challenge in automated *Varroa destructor* monitoring is the design of systems capable of processing complete hive board images in real-world environments. Despite progress in AI-based detection, the state of the art still lacks a robust solution for whole-board analysis. To manage computational and annotation complexity, most studies rely on analyzing subdivided regions rather than the full hive board image. Recent works utilizing high-resolution imaging continue to tile hive boards into frames or patches before applying AI-based detection. For instance, in [28], boards are segmented into quadrants, each captured and analyzed separately, thus limiting the scope to frame-level rather than true full-board inference. Similarly, in [22], high-resolution full-board images are split into smaller tiles, with each processed independently by a deep neural network. Studies such as [17] have made progress by assembling datasets of full-board images captured under realistic field conditions. While these images cover the entire board, detection is performed only on localized regions of interest (RoIs) identified through classical image processing techniques.

Table 7 summarizes relevant dataset characteristics, including image resolution and the average area of annotated *Varroa* mites (in pixels) for our data. For the other referenced datasets, the mean pixel size per *Varroa* was not reported, which limits the ability to directly compare object scale and detection difficulty across studies.

Future work could focus on approaches where multiple overlapping images are captured and subsequently stitched together to reconstruct a complete hive board image. The resulting composite can then be used as input for the detection model. However, this strategy may introduce new challenges, specifically in image alignment, maintaining lighting consistency across stitched regions, and increased computational demand for both reconstruction and analysis. In addition, our goal is to develop an integrated solution for beekeepers based on devices like smartphones or handheld devices. This approach ensures ease of use, eliminates the need for specialized equipment, and enables remote counting of *Varroa* mites, offering a practical solution for real-world needs.

Furthermore, explore the use of modern object detection architectures specifically optimized for small-object detection such as YOLOv8, EfficientDet, as these models may offer improved accuracy and computational efficiency, especially when training data are limited. In addition, incorporating adversarial-learning techniques may enhance model robustness under sparse annotations or variable environmental conditions. Transfer learning from large scale datasets could be critical for boosting performance in low-data regimes. Beyond single-task optimization, future pipelines may benefit from multi-objective AutoML frameworks that simultaneously optimize model architecture, hyperparameters, and inference latency. Frameworks such as DAICOX [29], GENESYS [30,31], and QuickCog [32] illustrate promising directions for adaptive, cognitive model search and deployment automation.

## 6. Conclusions

This study conducted a controlled comparison between a classical machine-learning pipeline (PCA + kNN + SVM) and deep-learning based object detection for *Varroa* mite detection in hyperspectral images. Despite the popularity of deep learning, our results show that a ML pipeline can outperform deeper architectures like Faster R-CNN when only a small number of annotated images are available Across all test images, the ML pipeline detected 25 of 26 *Varroa* mites (99.47 % recall), missing only a partially visible mite at the image edge. In contrast, Faster R-CNN with ResNet-50 detected 19 mites and ResNet-101 detected 20, and both DL models produced false positives by misclassifying debris, leading to counting errors. These absolute counts complement the percentage based metrics and underscore the practical advantage of the ML approach in low-data scenarios. Moreover, the ML pipeline runs entirely on a standard CPU—no GPU required and completes inference in under 10 ms, making it ideally suited for real-time deployment on mobile and edge hardware. By comparison, DL models learn rich features from data, but require substantially more training examples and specialized computation to achieve similar accuracy. Finally, the ML pipeline’s features and hyperparameters were manually tailored for *Varroa* detection, whereas the DL models expose many more degrees of freedom.

## Figures and Tables

**Figure 1 sensors-25-05075-f001:**
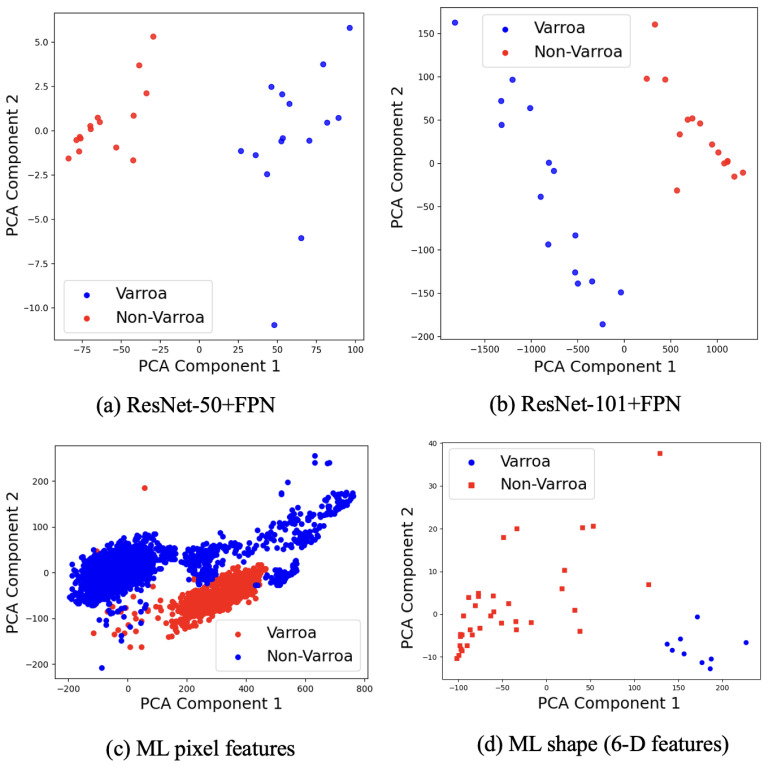
PCA projections of feature spaces.

**Figure 2 sensors-25-05075-f002:**
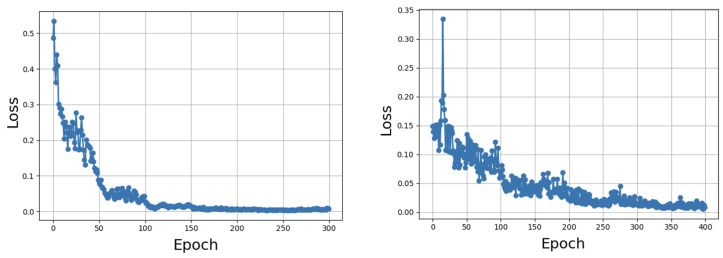
Training loss per epoch for (**left**) ResNet-50 + FPN and (**right**) ResNet-101 + FPN.

**Figure 3 sensors-25-05075-f003:**
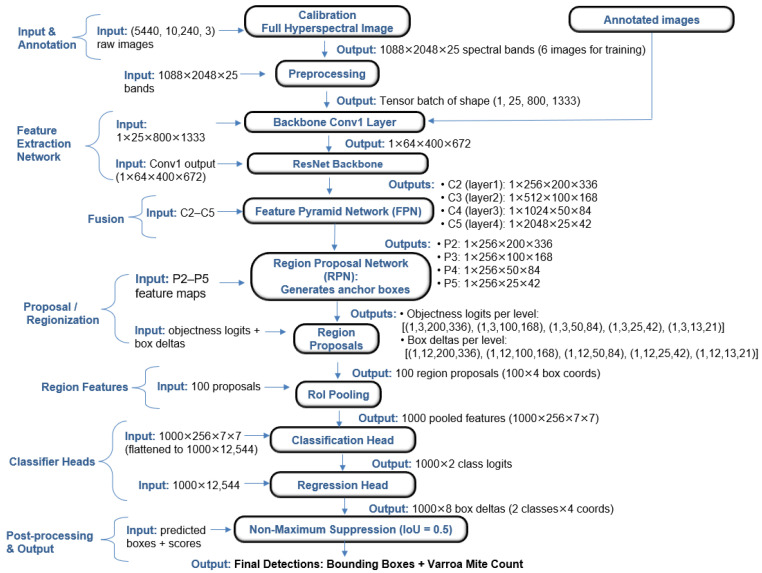
Faster R-CNN detection pipeline.

**Figure 4 sensors-25-05075-f004:**
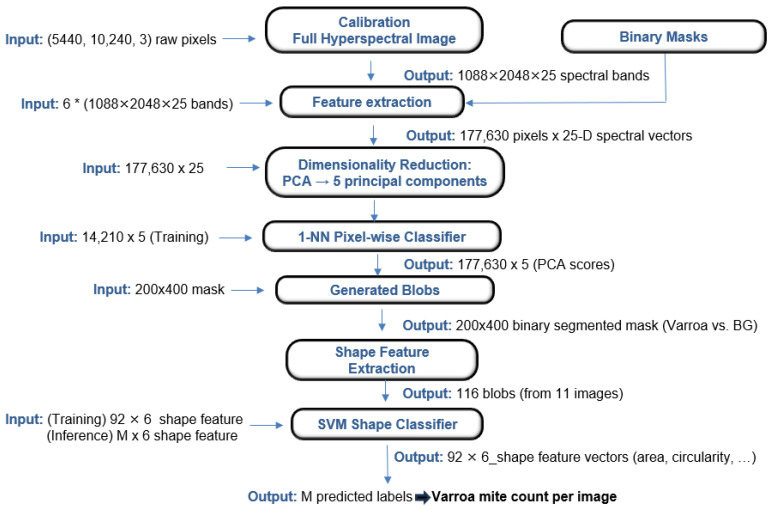
Overview of the machine-learning pipeline.

**Figure 5 sensors-25-05075-f005:**
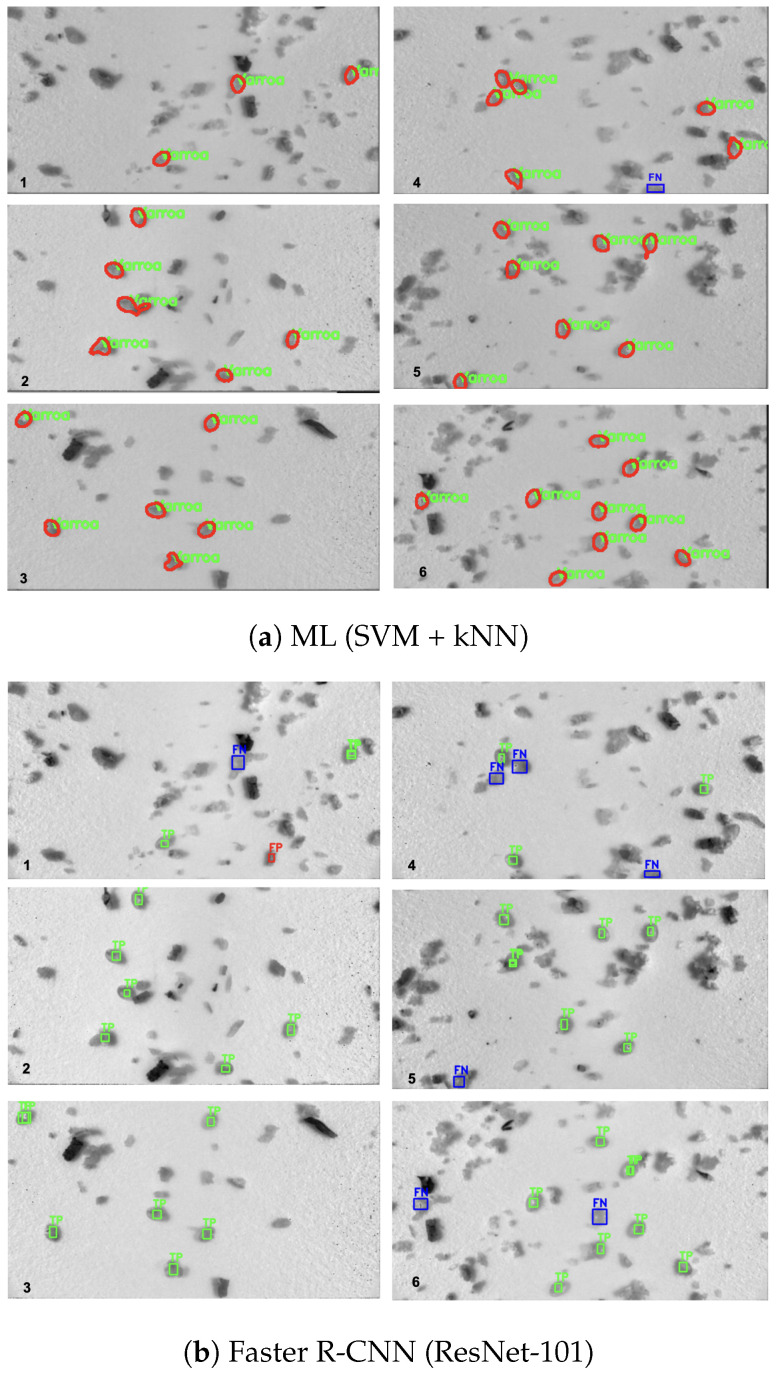
Detection outputs on six representative test images (GT counts: 3, 6, 6, 7, 7, 9 mites). Green = TP; red = FP; blue = FN.

**Figure 6 sensors-25-05075-f006:**
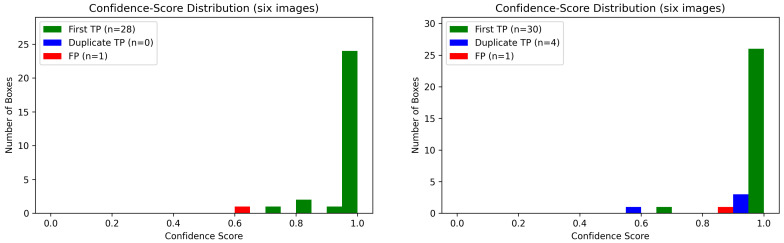
Confidence-score histograms at IoU = 0.50 for ResNet-50 (**left**) and ResNet-101 (**right**).

**Figure 7 sensors-25-05075-f007:**
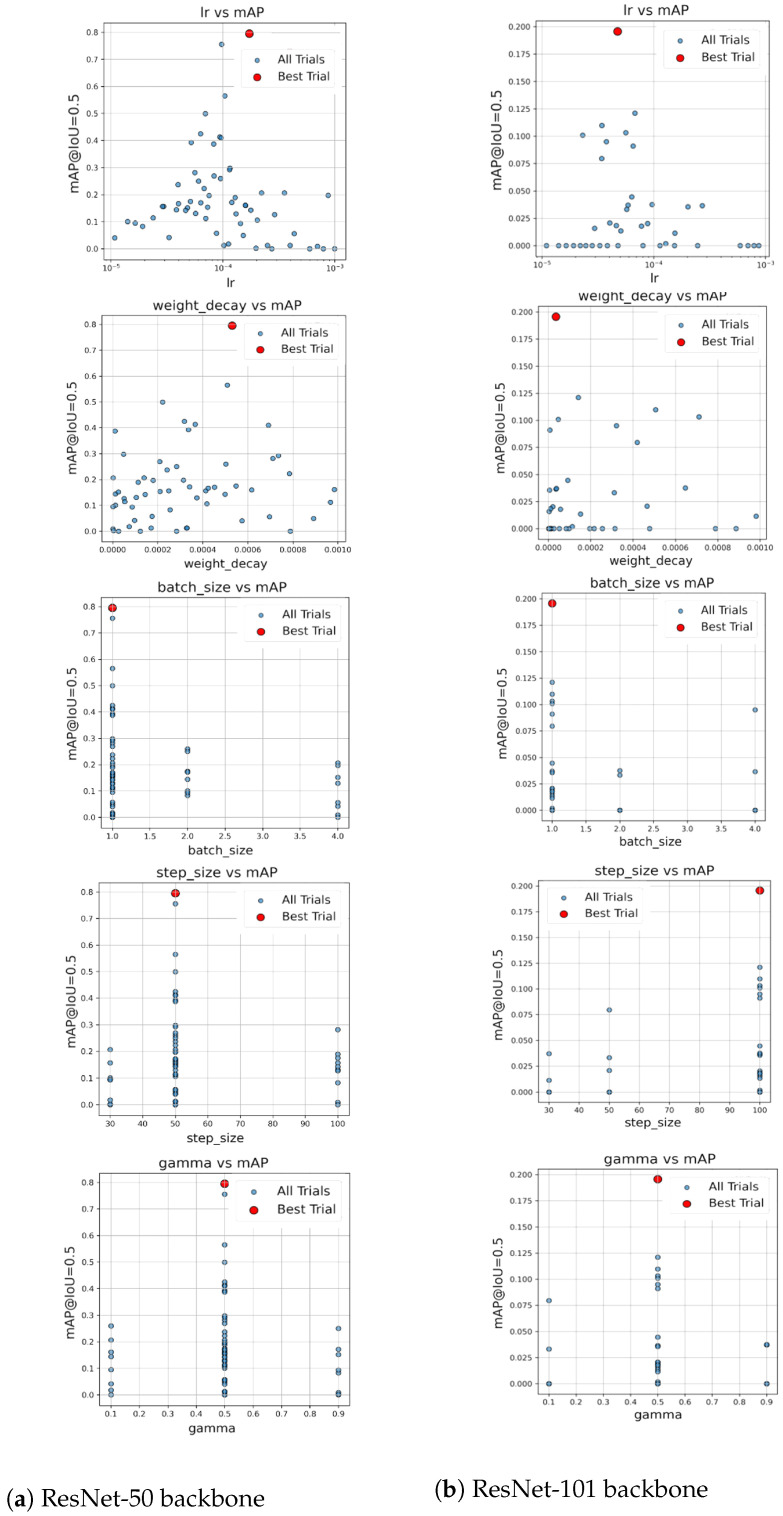
Relationship between hyperparameters and mAP for Faster R-CNN models.

**Figure 8 sensors-25-05075-f008:**
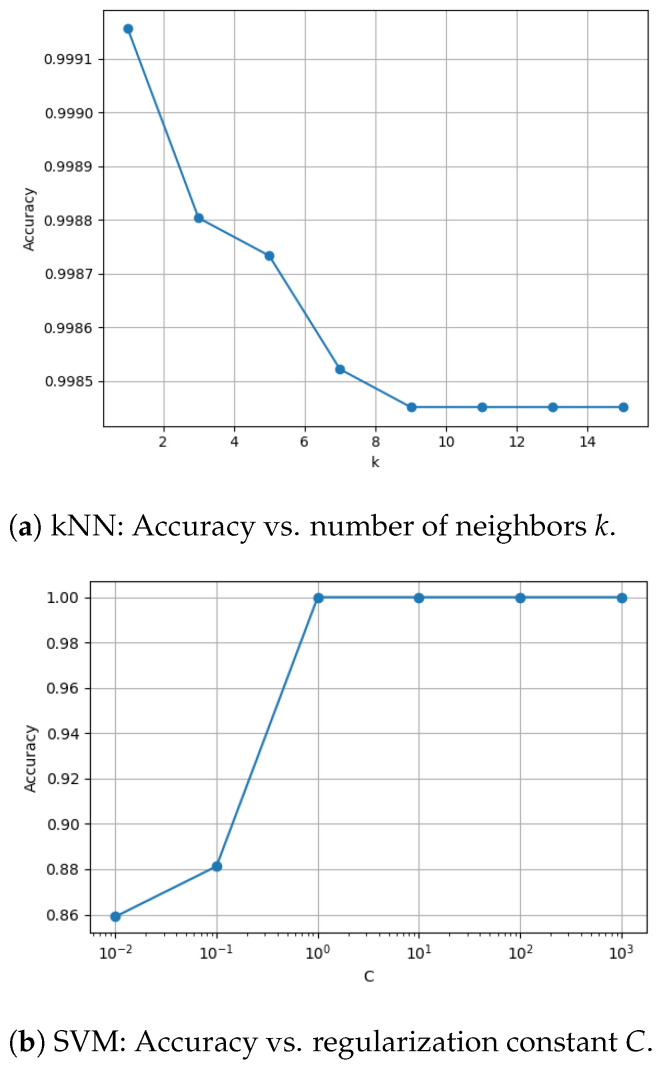
Effect of key ML hyperparameters on classification accuracy.

**Table 1 sensors-25-05075-t001:** Comparison of computational resources and execution time.

Method	Hardware Used	GPU Required	Training Time	Inference Time/ Image
ML (kNN + SVM)	Intel i7 CPU	No	∼68.7 s	∼0.0080 s
Faster R-CNN (ResNet-50)	NVIDIA Tesla V100-SXM2 (16 GB)	Yes	∼1043.7 s	∼0.6646 s
Faster R-CNN (ResNet-101)	NVIDIA Tesla V100-SXM2 (16 GB)	Yes	∼1055.3 s	∼0.6531 s

**Table 2 sensors-25-05075-t002:** Key hyperparameters for deep-learning models.

Model	Backbone	LR	Weight Decay	Scheduler
Faster R-CNN	ResNet-50 + FPN	$1.46\times 10^{-4}$	$5.31\times 10^{-4}$	StepLR (γ=0.5, step = 50)
Faster R-CNN	ResNet-101 + FPN	$4.74\times 10^{-5}$	$3.55\times 10^{-5}$	StepLR (γ=0.5, step = 100)

**Table 3 sensors-25-05075-t003:** Feature-spaces at every block for the ML and DL approaches.

Step	ML Pipeline	DL Pipeline
Input and annotation	1088 × 2048 × 25 bands + masks	1088 × 2048 × 25 bands + box & class labels
Feature extraction	Flatten pixels → N = 177,630 × 25-D spectral vectors	conv1→1 × 64 × 400 × 672, C2→1 × 256 × 200 × 336, C3→1 × 512 × 100 × 168, C4→1 × 1024 × 50 × 84, C5→1 × 2048 × 25 × 42
Feature selection/fusion	PCA on N × 25 → N × 5	FPN fuses C2–C5 → P2:1 × 256 × 200 × 336; P3:1 × 256 × 100 × 168; P4:1 × 256 × 50 × 84; P5:1 × 256 × 25 × 42
Proposal/ regionization	Segmented images → M = 116 blobs	Anchors → objectness logits + box deltas
Region features	Blobs → M × 6 shape-feature matrix	RoIAlign top-100 proposals → 1000 × 256 × 7 × 7 pooled features
Classifier heads	SVM on 92 × 6 (train)/M × 6 (test) → M labels	Cls-head on 1000 × (256 × 7 × 7) = 1000 × 12,544 → 1000 × 2 class logits Reg-head on 1000 × 12,544 → 1000 × 8 box offsets
Post-processing and output	Sum M “*Varroa*” labels → final count	NMS (IoU = 0.5) → final boxes + final count

**Table 4 sensors-25-05075-t004:** Quantitative comparison of machine-learning (ML) and deep-learning (DL) models for *Varroa* mite detection.

Model	Precision	Recall	F1-Score	TP/FP/FN
ML (SVM + kNN)	0.9983	0.9947	0.9918	37/0/1
Faster R-CNN (ResNet-50)	0.966	0.757	0.848	28/1/9
Faster R-CNN (ResNet-101)	0.971	0.829	0.894	30/5/7

**Table 5 sensors-25-05075-t005:** Performance metrics for ResNet-50 + FPN and ResNet-101 + FPN models.

Model	Precision (%)	Recall (%)	F1-Score (%)	mAP@0.5 (%)
ResNet-50 + FPN	71.0 ± 9.7	77.8 ± 7.0	76.7 ± 7.4	69.4 ± 10.0
ResNet-101 + FPN	67.4 ± 2.2	80.8 ± 1.7	73.5 ± 2.0	69.6 ± 0.7

**Table 6 sensors-25-05075-t006:** Per-image TP/FP/FN Counts for ML, ResNet-50, and ResNet-101 (Total GT = 38).

Image ID	GT Mites	ML (TP/FP/FN)	R50 (TP/FP/FN)	R101 (TP/FP/FN)
1	3	6/0/0	2/0/1	2/1/1 *
2	6	6/0/0	4/0/2	6/0/0
3	6	6/0/0	6/1/0	6/0/0 *
4	7	6/0/1	3/0/3	3/0/3
5	7	7/0/0	5/0/2	6/0/1 *
6	9	9/0/0	8/0/1	7/0/2 *

* In Images 1, 3, 5 and 6, ResNet-101 produced a duplicate TP (counted as detections).

**Table 7 sensors-25-05075-t007:** *Varroa* detection datasets and annotation characteristics.

Study	Resolution	Area/Frame (cm^2^)	*Varroa*/img	Mean *Varroa* Size (Pixels)	Notes
**Ours**	2048 × 1088 × 25	1.85 × 3.55	1–20	6000–9000	HSI, FRCNN
[22]	8064 × 6048	24 × 17.5	1–60	Not reported	Smartphone, FRCNN
[28]	48 MP/108 MP	23.5 × 18.5	1–351	Not reported	iPhone/Xiaomi, YOLOv11

## Data Availability

The datasets presented in this article are not readily available as the data are part of an ongoing study. Requests to access the datasets should be directed to the first author.

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
