# Peer review of "A Comparative Study of Hybrid Machine-Learning vs. Deep-Learning Approaches for Varroa Mite Detection and Counting"

_sensors, 2025, doi:10.3390/s25165075_

Round 1
Reviewer 1 Report
Comments and Suggestions for Authors
The manuscript presents a comparative analysis of machine learning (ML) and deep learning (DL) approaches for the detection of parasitic mites using spectral imaging. The introduction section contains a sufficient overview of the existing solutions based on RGB image acquisition. Due to the additional data provided by spectral imaging, the approach considered shows promise to improve detection accuracy in comparison with conventional RGB imaging. Since the state-of-the-art automated mite detection technique is not proposed and validated yet, the topic of the presented manuscript is highly relevant to the field.
Since the main considerations of the detection approach are published in the previous work, the novelty of the manuscript is limited and focused on the detection approach selection. The study described addresses the selection of optimal strategy with respect to the available training data amount and limited computational resources. Initially, DL models exhibited predictably lower performance in low-data scenarios. However, the comprehensive Discussion section thoroughly examines the results and their underlying causes, highlighting the primary limitations of ML and DL approaches applied to an identical dataset. Given the detailed descriptions of instrumentation, methods, and pipeline processes, the manuscript serves as a practical guide for automated target detection utilizing spectral imaging techniques.
Despite the absence of clear distinction between multi- and hyperspectral techniques and instrumentation, acquisition of spectral images with mosaic-like pattern of spectral filters embedded in image sensor (as well as multiaperture filtered camera approach) should be classified as a “multispectral imaging”. This is a suggestion from the reviewer and is not mandatory for revision.
In their previous work, the authors mentioned spatial scanning for panoramic image acquisition but have not demonstrated this feature. It appears that acquiring image arrays step-by-step across an entire hive board, with essential preliminary correction for illumination non-uniformity, could have addressed the low-data issue and enriched the comparison by investigating various dataset sizes. Such an analysis would assist the determination of the threshold dataset size for choosing between ML and DL approaches. Nevertheless, the current comparison sufficiently validates the key finding of ML’s superiority in low-data scenarios. The methodology used is correct and does not require substantial revision. The main conclusions are consistent with the metrics obtained presented in Results section and the critical analysis provided in Discussion section. The manuscript may be accepted following minor text-editing revisions.
Recommended Revisions:
- Please add the missing figure number in line 350.
- It is recommended to italicize Latin names throughout the manuscript.
- It is recommended to enlarge figure legend fonts.
Author Response
Dear Sir/Madam,
Please see the attachment
Best regards
Amira Ghezal

Reviewer 2 Report
Comments and Suggestions for Authors
I enjoyed reading the manuscript, it is quite amazing all the studies done for this Varroa mite mitigation.
Some figs, would benefit from slightly larger letters.
I have some minor comments (attachment)

Author Response
Dear Sir/Madam,
Please see the attachment.
Best regards
Amira Ghezal
